# Association between blood pressure categories and cardiovascular disease mortality in China

Jie Guo[1], Jun Lv[1,2,3], Yu Guo[4], Zheng Bian[4], Bang Zheng[1], Man Wu[1], Ling Yang[5], Yiping Chen[5], Jian Su[6], Jianqiang Zhang[7], Jvying Yao[8], Junshi Chen[9], Zhengming Chen[5], Canqing Yu[1,2]*, Liming Li[1,2]*, on behalf of the China Kadoorie Biobank Collaborative Group[¶]

1 Department of Epidemiology and Biostatistics, Peking University Health Science Center, Beijing, China, 2 Peking University Center for Public Health and Epidemic Preparedness & Response, Beijing, China, 3 Key Laboratory of Molecular Cardiovascular Sciences, Peking University, Beijing, China, 4 Chinese Academy of Medical Sciences, Beijing, China, 5 Nuffield Department of Population Health, Clinical Trial Service Unit & Epidemiological Studies Unit (CTSU), University of Oxford, Oxford, United Kingdom, 6 Jiangsu Center for Disease Control and Prevention, Nanjing, China, 7 Zhouquan Town Health Center, Tongxiang, China, 8 Gaoqiao Town Health Center, Tongxiang, China, 9 China National Center for Food Safety Risk Assessment, Beijing, China

¶ The members of the steering committee and collaborative group are listed in the acknowledgements
* yucanqing@pku.edu.cn (CY); lmleeph@vip.163.com (LL)

**Data Availability Statement:** Data cannot be shared publicly because there exist ethical restrictions. According to the Regulation of the People's Republic of China on the Administration of Human Genetic Resources, we are not allowed to

## Abstract

### Background

Blood pressure (BP) categories are useful to simplify preventions in public health, and diagnostic and treatment approaches in clinical practice. Updated evidence about the associations of BP categories with cardiovascular diseases (CVDs) and its subtypes is warranted.

### Methods and findings

About 0.5 million adults aged 30 to 79 years were recruited from 10 areas in China during 2004–2008. The present study included 430 977 participants without antihypertension treatment, cancer, or CVD at baseline. BP was measured at least twice in a single visit at baseline and CVD deaths during follow-up were collected via registries and the national health insurance databases. Multivariable Cox regression was used to estimate the associations between BP categories and CVD mortality.

Overall, 16.3% had prehypertension-low, 25.1% had prehypertension-high, 14.1% had isolated systolic hypertension (ISH), 1.9% had isolated diastolic hypertension (IDH), and 9.1% had systolic-diastolic hypertension (SDH). During a median 10-year follow-up, 9660 CVD deaths were documented. Compared with normal, the hazard ratios (95% CI) of prehypertension-low, prehypertension-high, ISH, IDH, SDH for CVD were 1.10 (1.01–1.19), 1.32 (1.23–1.42), 2.04 (1.91–2.19), 2.20 (1.85–2.61), and 3.81 (3.54–4.09), respectively. All hypertension subtypes were related to the increased risk of CVD subtypes, with a stronger association for hemorrhagic stroke than for ischemic heart disease. The associations were stronger in younger than older adults.

provide Chinese human clinical and genetic data abroad without an official approval. Researchers that are interested in accessing and analyzing data collected in the China Kadoorie Biobank (CKB) study may contact the data use and access committee (https://www.ckbiobank.org/site/Data +Access/Data+Access+Policy). As stated in the policy, as data custodian, the CKB study group must maintain the integrity of the database for future use and regulate data access to comply with prior conditions agreed with the Chinese government.

**Funding:** This study was supported by grants from the National Key R&D Program of China (http:// service.most.gov.cn/) to YG (2016YFC0900500, 2016YFC0900501) and CY (2016YFC0900504), and from the Chinese Ministry of Science and Technology (http://service.most.gov.cn/) to LL (2011BAI09B01), and from the National Natural Science Foundation of China (http://www.nsfc.gov. cn/) to CY (81973125) and LL (91846303, 91843302). The CKB baseline survey and the first re-survey were supported by a grant from the Kadoorie Charitable Foundation in Hong Kong. The funders had no role in study design, data collection and analysis, decision to publish, or preparation of the manuscript.

**Competing interests:** The authors have declared that no competing interests exist.

## Conclusions

Prehypertension-high should be considered in CVD primary prevention given its high prevalence and increased CVD risk. All hypertension subtypes were independently associated with CVD and its subtypes mortality, though the strength of associations varied substantially.

## Introduction

Hypertension is the most important risk factor for cardiovascular disease (CVD) [1]. According to single or combined elevations of systolic blood pressure (SBP) and diastolic blood pressure (DBP), hypertension is frequently classified into isolated systolic hypertension (ISH), isolated diastolic hypertension (IDH), and systolic-diastolic hypertension (SDH). The associations with CVD might vary among hypertension subtypes because of their different pathophysiological mechanisms [2,3]. Previous studies had demonstrated definite evidence for the CVD risk of ISH and SDH but the effect of IDH was inconclusive [4–6].

Moreover, a continuous and positive association with CVD had been demonstrated above SBP 115–120 mm Hg [7,8]. There is a growing concern about the CVD risk of SBP 120 mm Hg to 140 mm Hg. The Seventh Report of the Joint National Committee on the Prevention, Detection, Evaluation, and Treatment of High Blood Pressure (JNC-7) introduced a category as prehypertension with BP level of 120 to 139/80 to 89 mm Hg [9]. However, evidence about the CVD risk of prehypertension remains controversial [10,11]. A meta-analysis demonstrated that the increased CVD mortality was largely driven by the high-range of prehypertension [12]. In the 2017 American College of Cardiology/American Heart Association (ACC/AHA) BP guideline, SBP 130–139 and/or DBP 80–89 mm Hg was newly defined as "Hypertension stage 1" [13]. Current evidence found that "Hypertension stage 1" was associated with the increased CVD risk among the younger adults but studies conducted among the older did not find an increased risk, partly due to their small sample size of older adults [14–16]. According to the 2017 ACC/AHA guideline, both the prevalence of hypertension and the number of participants who should take antihypertension treatments increase dramatically [17]. Moreover, during the COVID-19 pandemic, individuals with CVD were more vulnerable to COVID-19 and had a greater risk of developing into a severe condition [18]. Therefore, to clarify the role of "Hypertension stage 1" on the development of CVD, especially among the older adults, is warranted for making prevention strategies about this BP group.

Besides, cerebrovascular diseases have been the top leading cause of mortality in China [19]. Moreover, hemorrhagic stroke accounted for a larger proportion in the Chinese population than western populations [20] and the mortality from hemorrhagic stroke was higher than that from ischemic stroke [21]. However, there is limited evidence for the associations of BP categories with the major subtypes of cardiovascular disease (i.e., ischemic heart disease, ischemic stroke, and hemorrhagic stroke) in a Chinese population. We hypothesized that the newly defined hypertension was associated with increased CVD risk across a wide range of age, and the strength of the association of different hypertension subtypes with CVD and its subtypes might vary. We aimed to provide more detailed information about the associations of BP categories with overall and specific CVD mortality based on the China Kadoorie Biobank (CKB) study.

## Materials and methods

### Study population

Details of the CKB study design and methods have been reported elsewhere [22,23]. The CKB is a community-based prospective cohort study, involving over 0.5 million adults from 10 areas of China between 2004 and 2008. All men and women aged 30–79 years who were permanently resident and without major disability in each administrative unit were identified and invited to participate [22]. Ethics approvals were obtained from the Ethical Review Committee of Oxford University, the China National Center for Disease Control and Prevention (CDC), and from institutional research boards at the local CDCs in the 10 regions, and all participants provided written informed consent. The study was in accordance with the Declaration of Helsinki.

For the current study, we excluded participants with missing data for covariates (n = 49), or with implausible censoring date (n = 1), or with CVD or cancer at baseline (n = 25 511). Moreover, we further excluded participants with the antihypertension treatment at baseline (n = 65 168), because both dose and types of antihypertensive medications may influence BP levels and lead to misclassification of BP categories. Finally, we included 430 977 participants in the main analyses. By December 31, 2016, 4434 (1.03%) participants were lost to follow-up.

### Assessment of BP categories

BP was measured twice by trained staff using a UA-779 digital monitor after they remained at rest in the seated position for at least 5 minutes [24]. If the difference between the two measurements was >10 mm Hg for SBP, a third measurement was required. Only the last two readings were recorded and used to calculate the average of SBP and DBP [7].

According to the JNC-7, BP categories were defined into five groups 1) normal (SBP <120 and DBP <80 mm Hg); 2) prehypertension (SBP 120–139 and/or DBP 80–89 mm Hg); 3) ISH (SBP ≥140 and DBP <90 mm Hg); 4) IDH (SBP <140 and DBP ≥90 mm Hg); 5) SDH (SBP ≥140 and DBP ≥90 mm Hg) [9]. In the 2017 ACC/AHA hypertension guideline, hypertension was defined as SBP ≥130 mmHg and/or DPB ≥90 mmHg [13]. To estimate the effect of "Elevated" and "Hypertension stage 1" in the 2017 ACC/AHA hypertension guideline, we further divided prehypertension into prehypertension-low (equal to "Elevated", SBP 120–129 and DBP <80 mm Hg) and prehypertension-high (equal to "Hypertension stage 1", SBP 130–139 and/or DBP 80–89 mm Hg) [13].

### Assessment of covariates

Potential confounding variables in this study included sociodemographic characteristics (age, sex, education, and marital status), lifestyle behaviors (tobacco smoking, alcohol consumption, physical activity, and consumption of red meat, fresh fruits, and vegetables), diabetes, menopausal status for female only, and family medical history (heart attack, stroke). Physical activity was calculated by multiplying the metabolic equivalent tasks (METs) value for a particular type of physical activity by hours spent on that activity per day and summing the MET-hours for all activities for each day.

Height, body weight, and heart rate were measured by trained staff. Body mass index (BMI) was calculated as weight in kilograms divided by height in meters squared. Prevalent diabetes was defined as a measured fasting blood glucose concentration of ≥7.0 mmol/L, a measured random blood glucose concentration of ≥11.1 mmol/L, or self-reported diagnoses of diabetes.

## Assessment of outcomes

The vital status of participants was collected through linkage with regional disease and death registries, and with the new national health insurance databases. To minimize the under-reporting of deaths, we also carried out active follow-up annually, by reviewing residential records, visiting local communities, or directly contacting participants [23].

All deaths were coded using the 10th International Classification of Diseases (ICD-10). The main outcome measures in our analysis were mortality from CVD (ICD-10 codes I00 to I99), ischemic heart disease (I20 to I25), myocardial infarction (I21 to I23), cerebrovascular disease (I60 to I69), hemorrhagic stroke (I61), and ischemic stroke (I63).

## Statistical analysis

Baseline characteristics were described as means and standard errors or percentages in each BP category, with adjustment for age, sex, and survey sites as appropriate, using either multiple linear regression (for continuous variables) or logistic regression (for categorical variables).

Person-years at risk were calculated from the baseline to the date of death, loss to follow-up, or 31 December 2016, whichever occurred first. Cox proportional hazard models, stratified by age at risk (in 5-year intervals), sex, and survey sites (10 regions), were used to estimate the hazard ratios (HR) and 95% confidence intervals (CI) for CVD mortality related to BP categories, with age as the timescale. The proportional hazards assumption was checked using the Schoenfield residuals, and no violation was observed. The associations of BP categories with mortality from total CVD and its subtypes were assessed after adjustment for age (continuous) at recruitment; and subsequently adjusted for the level of education (no formal or primary school, middle school or high school, or college or higher), marital status (married, or others), smoking status (5 categories: never or occasional smoker, ex-smoker who quit not because of illness, ex-smoker who quit because of illness and current smoker divided into 3 groups according to cigarette equivalents/day [<15, 15−25, ≥25]), alcohol consumption (7 categories: never or occasional or seasonal drinker, ex-regular drinker of reduced intake, 1 to 5 days/week, almost daily drinker divided into 4 groups according to total grams of alcohol [<15, 15−<30, 30−<60, ≥60]), intake frequencies of red meat, fresh fruits, and vegetables (daily, 4−6 days/week, 1−3 days/week, or monthly, rarely or never), physical activity in MET-hours a day (continuous), survey seasons (spring; summer: June, July, August; autumn; winter: December, January, February), menopausal status (for female only, postmenopausal or others); and finally further adjustment for prevalent diabetes at baseline, family medical history, BMI (continuous), and heart rate (continuous). The Nelson-Aalen method was used to describe the cumulative hazard of CVDs during the follow-up across BP categories.

Moreover, we conducted subgroup analyses by age (30−49, 50−59, or 60−79 years), sex (male or female), and survey sites (10 areas) and the interaction effect was estimated by adding a cross-product term (e.g., age groups × BP categories) to the Cox model. In sensitivity analyses, we 1) investigated the associations between BP categories and CVD mortality after excluding the first 2 years of follow-up; or excluding those who had diabetes at baseline; 2) compared the risk of ISH, IDH, and SDH with further controlling SBP or DBP by conducting analyses in the range of SBP/DBP level of 140−159/90−99 mm Hg and of ≥160/100 mm Hg separately.

Statistical analyses were performed using SAS 9.3 (SAS Institute, Cary, NC). All *P* values were two-sided, and we defined statistical significance as *P* <0.05.

## Results

### Baseline characteristics of participants by BP categories

Among 430 977 participants, the mean age was 50.6 years, 58.7% were female, and 57.6% were from rural areas. Overall, 16.3% had prehypertension-low, 25.1% had prehypertension-high,

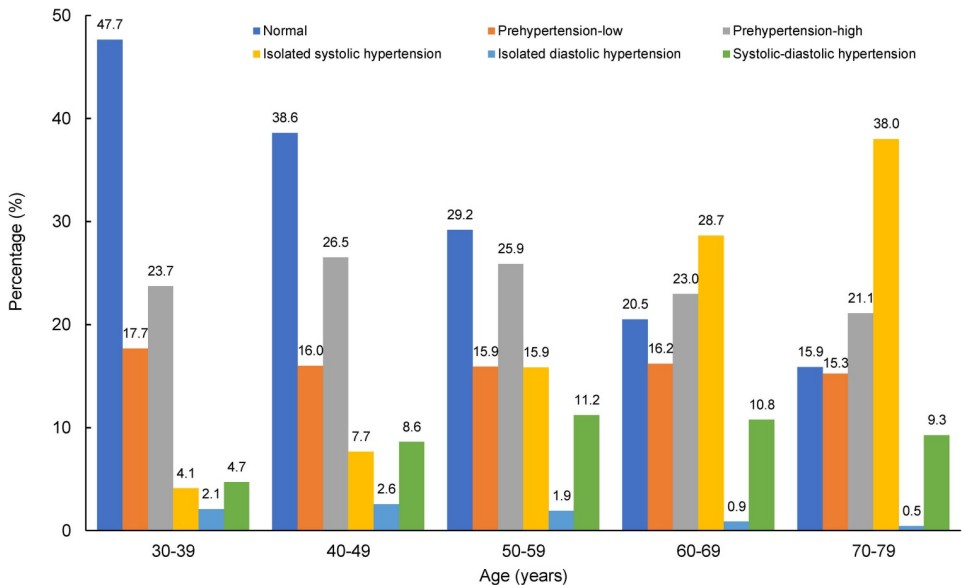

**Fig 1. Percentages of blood pressure categories across different age groups.**

14.1% had ISH, 1.9% had IDH, and 9.1% had SDH. The prevalence of ISH in older participants was higher than that in younger, while the opposite was observed for IDH (**Fig 1**).

Compared with normal BP, participants with prehypertension-low, prehypertension-high, or hypertension subtypes were older (except for IDH $P = 0.10$) and were less likely to be female, had a lower level of education (except for IDH $P = 0.99$), were more likely to live in rural areas (except for IDH $P = 0.32$), were less likely to smoke but more likely to drink regularly, were more likely to consume red meat (except for prehypertension-low $P = 0.33$) but less likely to consume fresh fruit, had a higher level of BMI and heart rate, and had a higher prevalence of diabetes (**Table 1**). The distribution of baseline characteristics of the study population without adjustment for age, sex, and survey sites was presented in **S1 Table**.

## Association of BP categories with CVD mortality

During 4.3 million person-years of follow-up (mean duration of follow-up: 10.0 years; median 10.2 years), there were 9660 deaths from CVD, 3564 from ischemic heart diseases (including 2248 myocardial infarction), and 5168 from cerebrovascular diseases (including 3092 hemorrhagic strokes and 965 ischemic strokes).

With normal BP as the reference, prehypertension had a higher risk of overall CVD, cerebrovascular disease, and hemorrhagic stroke (**S2 Table**). Similar patterns were also observed for prehypertension-low and prehypertension-high (**Table 2**). Besides, prehypertension-high was related to the increased risk of ischemic heart disease and ischemic stroke. All hypertension subtypes were associated with the increased mortality of overall CVD and its subtypes, after basic- or multi-adjustment for potential confounding factors (**S3 Table**). The multi-adjusted HRs for overall CVD were highest for SDH (adjusted HR, 3.81 [95%CI, 3.54 to 4.09]), followed by IDH (2.20 [95%CI, 1.85 to 2.61]), ISH (2.04 [95%CI, 1.91 to 2.19]), prehypertension-high (1.32 [95%CI, 1.23 to 1.42]), and finally prehypertension-low (1.10 [95%CI, 1.01 to 1.19]) (**Table 2**). The Nelson-Aalen curves of the cumulative hazard of CVDs visually showed that SDH had the highest hazard curve (**S1 Fig**).

**Table 1. Baseline characteristics of the study population by baseline BP categories[a].**

| Characteristic | Normal | Prehypertension-low | Prehypertension-high | Hypertension | | |
| --- | --- | --- | --- | --- | --- | --- |
| | | | | ISH | IDH | SDH |
| Total, No. | 144 765 | 70 130 | 107 960 | 60 708 | 8387 | 39 027 |
| Age, mean (SE), y | 47.7 (0.03) | 50.5 (0.04) | 50.5 (0.03) | 58.0 (0.04) | 47.9 (0.11) | 53.0 (0.05) |
| Female, No. (%) | 96 918 (66.1) | 39 937 (56.9) | 58 329 (54.0) | 35 134 (60.2) | 3794 (44.7) | 18 784 (49.0) |
| Education level, No. (%) | | | | | | |
| No formal education or primary | 59 294 (48.5) | 35 191 (49.8) | 53 136 (49.2) | 40 644 (51.4) | 3246 (48.6) | 21 588 (50.6) |
| Middle or high school | 73 853 (45.0) | 31 459 (44.7) | 49 166 (45.4) | 18 249 (43.6) | 4442 (45.6) | 15 814 (44.9) |
| College or higher | 11 618 (6.5) | 3480 (5.5) | 5658 (5.5) | 1815 (5.0) | 699 (5.8) | 1625 (4.5) |
| Rural area, No. (%) | 73 840 (50.1) | 42 291 (60.2) | 65 486 (60.6) | 37 856 (64.4) | 4324 (50.6) | 24 396 (63.1) |
| Married, No. (%) | 134 038 (91.3) | 64 446 (91.8) | 99 571 (91.7) | 52 636 (91.1) | 7849 (91.1) | 35 419 (91.0) |
| Regular smoking, No. (%) | 33 951 (29.0) | 20 331 (27.7) | 32 518 (27.0) | 16 082 (26.3) | 2917 (26.8) | 13 142 (27.2) |
| Male | 31 785 (66.5) | 19 323 (63.6) | 31 197 (62.4) | 15147 (61.3) | 2852 (61.8) | 12689 (63.1) |
| Female | 2166 (3.1) | 1008 (2.4) | 1321 (2.2) | 935 (1.9) | 65 (2.3) | 453 (2.2) |
| Regular alcohol intake, No. (%) | 15 732 (13.5) | 10 150 (14.1) | 19 655 (16.0) | 9628 (16.6) | 2196 (18.7) | 9724 (20.1) |
| Male | 13 635 (29.5) | 9330 (31.4) | 18 320 (35.6) | 8875 (37.2) | 2085 (40.8) | 9239 (44.4) |
| Female | 2097 (2.3) | 820 (2.0) | 1335 (2.2) | 753 (2.1) | 111 (3.1) | 485 (2.7) |
| Average weekly consumption, mean (SE), day/week [b] | | | | | | |
| Fresh vegetables | 6.82 (0.002) | 6.82 (0.002) | 6.82 (0.002) | 6.82 (0.003) | 6.81 (0.008) | 6.81 (0.004) |
| Fresh fruits | 2.59 (0.006) | 2.48 (0.008) | 2.48 (0.007) | 2.37 (0.009) | 2.44 (0.024) | 2.30 (0.011) |
| Red meat | 3.90 (0.006) | 3.91 (0.008) | 3.96 (0.006) | 3.96 (0.009) | 3.99 (0.023) | 3.94 (0.010) |
| Postmenopausal, No. (%) [c] | 33 002 (49.7) | 18 865 (48.5) | 27 871 (49.1) | 26 430 (48.0) | 1447 (50.5) | 11 023 (49.1) |
| Physical activity, mean (SE), MET- hr/day | 21.9 (0.03) | 22.3 (0.05) | 21.9 (0.04) | 21.8 (0.05) | 21.1 (0.13) | 21.6 (0.06) |
| Heart rate, mean (SE), bpm | 76.2 (0.03) | 77.7 (0.04) | 79.7 (0.03) | 79.8 (0.05) | 82.9 (0.12) | 83.1 (0.06) |
| Body mass index, mean (SE), kg/m$^2$ | 22.4 (0.01) | 23.3 (0.01) | 23.8 (0.01) | 24.3 (0.01) | 24.4 (0.03) | 24.9 (0.02) |
| Diabetes, No. (%) | 3574 (2.6) | 2635 (3.9) | 4847 (4.7) | 5177 (6.9) | 366 (4.7) | 2409 (6.0) |
| Family medical history, No. (%) | 25 525 (17.2) | 12 555 (18.2) | 21 193 (19.7) | 11 662 (20.2) | 2012 (22.3) | 9347 (23.9) |
| Heart attack | 4707 (3.0) | 2078 (3.1) | 3314 (3.1) | 1613 (3.1) | 338 (3.5) | 1329 (3.5) |
| Stroke | 21 875 (14.9) | 10 918 (15.8) | 18 684 (17.3) | 10 445 (17.8) | 1772 (19.8) | 8390 (21.3) |
| SBP, mean (SE), mmHg | 109.7 (0.02) | 124.3 (0.03) | 129.8 (0.03) | 150.0 (0.04) | 133.5 (0.09) | 161.8 (0.04) |
| DBP, mean (SE), mmHg | 68.0 (0.02) | 72.4 (0.02) | 80.1 (0.02) | 81.4 (0.03) | 92.1 (0.07) | 97.6 (0.03) |

Abbreviations: BP, blood pressure; SE, standard error; MET, metabolic equivalent of task; bpm, beat per minute.

[a] All variables were adjusted for age, sex, and survey sites when appropriate.

[b] Average weekly consumptions of red meat, fresh vegetables, and fruits were calculated by assigning participants to the midpoint of their consumption category.

[c] Only for female.

## Subgroup and sensitivity analyses

Prehypertension-high and all hypertension subtypes were associated with increased CVD mortality among all age groups (**Fig 2**). Moreover, we observed stronger associations in younger participants than older ones (all *P* values for interaction <0.005, except for ischemic stroke *P* = 0.20). Heterogeneity by sex was observed for overall CVD, ischemic heart disease, cerebrovascular diseases, and hemorrhagic stroke (**S4 Table**). **S5 Table** shows the associations between BP categories and CVD mortality across 10 survey sites. There was a statistically significant interaction between BP categories and survey sites (*P* for interaction = 0.016).

The associations between BP categories and CVD mortality were not materially altered after excluding the first 2 years of follow-up (**S6 Table**) or excluding participants who had diabetes at baseline (**S7 Table**). In hypertension, the multivariable-adjusted HR of SDH for CVD

**Table 2. Associations of BP categories with mortality from CVDs and its major subtypes among 430 977 participants.**

| Cause of death | Normal | Prehypertension-low | Prehypertension-high | Hypertension | | |
| --- | --- | --- | --- | --- | --- | --- |
| | | | | ISH | IDH | SDH |
| No. of participants | 144 765 | 70 130 | 107 960 | 60 708 | 8387 | 39 027 |
| No. of person-years | 1 462 055 | 705 310 | 1 082 972 | 593 913 | 85 056 | 383 785 |
| Cardiovascular disease | | | | | | |
| No. of deaths | 1330 | 1039 | 1787 | 3133 | 148 | 2223 |
| Incidence rate (no./1,000 person-y) | 0.91 | 1.47 | 1.65 | 5.28 | 1.74 | 5.79 |
| HR (95%CI) | Reference | 1.10 (1.01–1.19) | 1.32 (1.23–1.42) | 2.04 (1.91–2.19) | 2.20 (1.85–2.61) | 3.81 (3.54–4.09) |
| Ischemic heart disease | | | | | | |
| No. of deaths | 586 | 423 | 691 | 1159 | 56 | 649 |
| Incidence rate (no./1,000 person-y) | 0.40 | 0.60 | 0.64 | 1.95 | 0.66 | 1.69 |
| HR (95%CI) | Reference | 1.00 (0.88–1.13) | 1.14 (1.02–1.28) | 1.66 (1.50–1.85) | 1.76 (1.32–2.30) | 2.48 (2.21–2.80) |
| Myocardial infarction | | | | | | |
| No. of deaths | 378 | 258 | 444 | 713 | 32 | 423 |
| Incidence rate (no./1,000 person-y) | 0.26 | 0.37 | 0.41 | 1.20 | 0.38 | 1.10 |
| HR (95%CI) | Reference | 0.94 (0.80–1.11) | 1.14 (0.99–1.31) | 1.65 (1.44–1.88) | 1.58 (1.08–2.24) | 2.45 (2.11–2.84) |
| Cerebrovascular disease | | | | | | |
| No. of deaths | 577 | 499 | 900 | 1683 | 70 | 1439 |
| Incidence rate (no./1,000 person-y) | 0.39 | 0.71 | 0.83 | 2.83 | 0.82 | 3.75 |
| HR (95%CI) | Reference | 1.20 (1.06–1.36) | 1.53 (1.37–1.70) | 2.52 (2.28–2.78) | 2.51 (1.94–3.21) | 5.60 (5.06–6.21) |
| Hemorrhagic stroke | | | | | | |
| No. of deaths | 315 | 279 | 530 | 962 | 45 | 961 |
| Incidence rate (no./1,000 person-y) | 0.22 | 0.40 | 0.49 | 1.62 | 0.53 | 2.50 |
| HR (95%CI) | Reference | 1.26 (1.07–1.48) | 1.69 (1.47–1.95) | 2.90 (2.53–3.32) | 2.86 (2.06–3.88) | 6.91 (6.05–7.92) |
| Ischemic stroke | | | | | | |
| No. of deaths | 126 | 93 | 181 | 331 | 13 | 221 |
| Incidence rate (no./1,000 person-y) | 0.09 | 0.13 | 0.17 | 0.56 | 0.15 | 0.58 |
| HR (95%CI) | Reference | 1.01 (0.77–1.33) | 1.38 (1.10–1.75) | 2.04 (1.65–2.55) | 2.13 (1.14–3.65) | 3.93 (3.12–4.97) |

Abbreviations: BP, blood pressure; CVDs, cardiovascular diseases; ISH, isolated systolic hypertension; IDH, isolated diastolic hypertension; SDH, systolic-diastolic hypertension; HR, hazard ratio; CI, confidence interval.

Multi-adjusted hazard ratios were adjusted for age (continuous variable); education level (no formal or primary school, middle school or high school, or college or higher); marital status (married, others); smoking status (never or occasional smoker, ex-smoker who quit not because of illness, ex-smoker who quit because of illness and current smoker divided into 3 groups according to cigarette equivalents/day [<15, 15–25, ≥25]); alcohol consumption (never or occasional or seasonal drinker, ex-regular drinker of reduced intake, 1 to 5 days/week, almost daily drinker divided into 4 groups according to total grams of alcohol [<15, 15-<30, 30-<60, ≥60]); intake frequencies of vegetables, fruits, and red meat (daily, 4 to 6 days/per week, 1 to 3 days/per week, or monthly or rarely/never); physical activity (continuous variable); body mass index (BMI, continuous variable); survey season (spring; summer: June, July, August; autumn; winter: December, January, February); heart rate (continuous variable); diabetes at baseline (2 categories, presence or absence); family history of heart attack, or stroke (2 categories, presence or absence, only adjusted for in corresponding analysis of cause-specific mortality), and were stratified according to age at risk (in 5-year intervals), sex, and survey sites.

mortality was higher than that of ISH and IDH (*P* <0.001), while the effect of ISH on CVD mortality was almost similar to that of IDH (S2 Fig).

## Discussion

The present study, involving more than 0.4M people living in China, found that both prehypertension-low and prehypertension-high were associated with higher CVD mortality independent of other cardiovascular risk factors. All hypertension subtypes were associated with increased mortality from overall CVD and CVD subtypes, and the CVD risk of SDH was

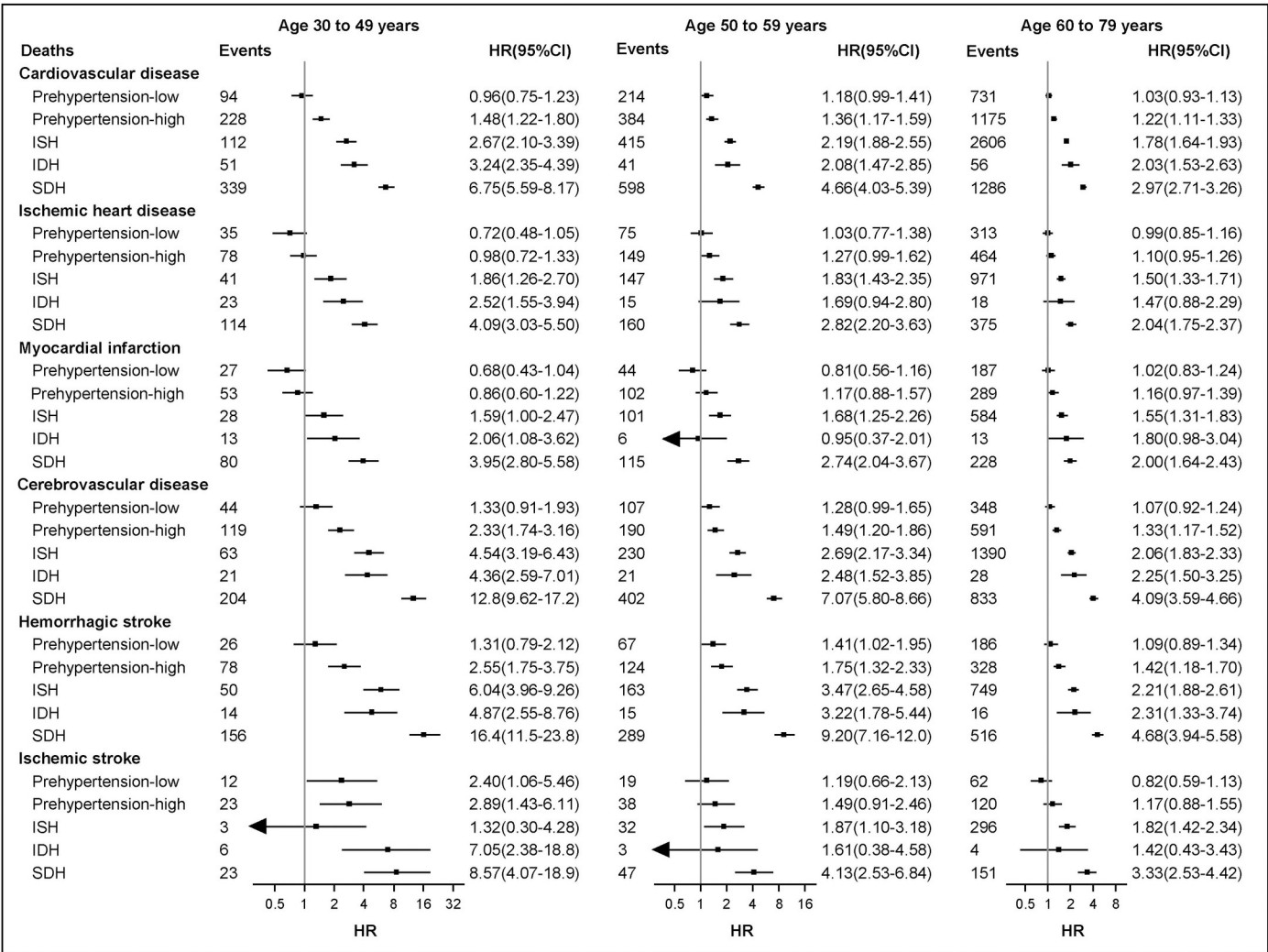

**Fig 2. Associations of BP categories with mortality from CVDs and its major subtypes by age groups.** Abbreviations: BP, blood pressure; CVDs, cardiovascular diseases; HR, hazard ratios; CI, confidence interval. Reference was normal BP. Multi-adjusted HR were adjusted for age, education level, marital status, smoking status, alcohol consumption, intake frequencies of vegetables, fruits, and red meat, physical activity, body mass index, survey season, heart rate, diabetes at baseline, family history of heart attack, stroke (only adjusted for in corresponding analysis of cause-specific mortality) and were stratified according to sex and survey sites. Statistically significant heterogeneity was observed in the associations between blood pressure categories and CVD mortality across age groups (all *P* values for interaction <0.005, except for ischemic stroke *P* = 0.20). Data markers represent the point estimate of hazard ratio. Error bars represent 95% confidence interval. Black arrows represent the confidence intervals exceed the X-axis value.

higher than that of ISH and IDH. The associations between BP categories and cerebrovascular diseases were stronger than for ischemic heart diseases. Furthermore, the associations between BP categories and CVD mortality were stronger in younger participants than in older ones.

The 2017 ACC/AHA hypertension guideline defined the 130-139/80-89 mm Hg as "Hypertension stage 1" [13]. Based on the guideline, about a quarter of participants (i.e., prehypertension-high) in our study would be newly defined as hypertension. Previous studies found an increased CVD risk of this BP category in younger adults, which was consistent with our results [15,25]. However, the findings were mixed in older adults. Some studies reported that this BP category was not associated with the increased CVD risk [15,25], while others reported a slightly higher CVD mortality for those aged ≥65 years (HR [95%CI]: 1.22 [1.04 to 1.44]) [26]. In our study, the CVD risk of this BP category became weaker in participants aged 60 to

79 years, but there was still a 22% higher CVD mortality than the normal BP. Moreover, this BP category was associated with increased mortality from CVD subtypes, especially for hemorrhagic stroke, with a 69% higher risk compared to normal BP. Given the high prevalence of cerebrovascular diseases, especially hemorrhagic stroke in China [20], managing the BP of that newly defined hypertension in the public health practice may benefit for reducing the CVD burden.

Previous studies have reported consistent associations of both ISH and SDH with the CVD mortality [4,6,27], and clinical trial also detected that individuals with ISH or SDH could benefit from antihypertensive treatment [28], but whether IDH was associated with an increased CVD risk was still controversial [4,5,29]. In the current study, all hypertension subtypes were related to the increased mortality from CVD and specific CVDs, and IDH was at least as important as ISH in predicting future CVD mortality. These results also supported current guidelines which recommended pharmacologic treatments based on DBP as well as SBP [9,30]. In line with previous studies [4,6], the present study demonstrated that SDH conferred the highest CVD risk, followed by IDH and ISH. The heterogeneity among hypertension subtypes might be explained by the higher mean level of SBP or DBP of SDH than that of ISH or IDH. However, after controlling for SBP or DBP, the CVD risk of SDH was still higher than that of ISH. Similar results were also observed for cerebrovascular disease, hemorrhagic stroke, and ischemic stroke. Our finding suggested that incorporating SBP and DBP might improve the prediction of CVD risk models, and future studies are needed to clarify potential mechanisms for the heterogeneity among hypertension subtypes. Likewise, we also identified that the strength of the associations between hypertension subtypes and specific CVD mortality varied significantly, with stronger for cerebrovascular disease than for ischemic heart disease, and extreme for hemorrhagic stroke than for ischemic stroke.

Previous literature found that the associations between BP categories and CVD deaths varied by age, with a stronger association in younger adults than older ones [6,31]. Consistent with these findings, we also identified stronger associations between BP categories and major CVDs (except ischemic stroke) among younger compared with older. Additionally, the mechanisms of ISH in young are still unclear. Some studies showed that in young and middle-aged adults, ISH was "pseudo" or "spurious" hypertension attributed to amplification of central aortic waveform, while others found ISH might be related to increased stroke volume and aortic stiffness [32–34]. However, the increased CVD risk of ISH in the current study indicated the terms "pseudo" or "spurious" hypertension might be unjustified.

To the best of our knowledge, this is the largest prospective study to investigate the associations of BP categories with mortality from CVD and its major subtypes in a Chinese population. The chief strengths of this study included the unified standard methods for collecting information, the sufficient number of CVD outcomes, and the rigorous check of diagnosis of CVD. This study also had several limitations. First, BP was obtained in a single baseline survey, without considering the fluctuation of BP in the daytime, which may misclassify individuals with "white-coat hypertension" or "masked hypertension" [35,36]. However, it is not feasible to monitor ambulatory BP for each participant in a large-scale population study. Second, we excluded participants who taking antihypertensive medicines, which might cause selection bias and limit the extrapolation of our findings. However, antihypertensive medication would affect the patients' blood pressure level, leading to misclassification of BP categories. Hence, it is more reasonable to restrict study participants without antihypertensive treatment. Third, elevated low-density lipoprotein (or total) cholesterol, a risk factor for CVD [37], was not available in the present study. A meta-analysis reported that total cholesterol (TC) was positively associated with ischemic heart disease mortality, but there was rather a weak association of TC with mortality from cerebrovascular disease and hemorrhagic stroke [38]. Therefore,

failing to adjust for TC is unlikely to have any appreciable impact on the associations with cerebrovascular disease in our study. Fourth, under-reporting of cardiovascular deaths might have occurred during follow-up, but the probability of under-reported would not depend on BP categories, and we also used multiple ways to minimize the under-reporting of deaths.

## Conclusion

The definition of hypertension is one of the most notable changes in the 2017 ACC/AHA hypertension guideline. The present study provided important evidence about the long-term CVD risk of those new hypertensives (i.e., "Hypertension stage 1" in 2017 guideline and prehypertension-high in the current study) and highlighted its important role in CVD primary prevention, both due to the high prevalence and be associated with higher CVD mortality. All hypertension subtypes were related to the increased mortality from CVDs, especially from hemorrhagic strokes, and should be considered in BP management regardless of age and gender.

## Supporting information

**S1 Checklist. STROBE statement for observational studies.**
(DOCX)

**S1 Fig. Nelson-Aalen cumulative hazard for cardiovascular diseases according to the blood pressure categories.**
(DOCX)

**S2 Fig. Associations of ISH, IDH, and SDH with mortality from CVDs and its major subtypes in stage 1 hypertension, stage 2 hypertension and total hypertension.**
(DOCX)

**S1 Table. Baseline characteristics of the study population by baseline BP categories.**
(DOCX)

**S2 Table. Associations of blood pressure categories with cardiovascular diseases mortality among 430 977 participants.**
(DOCX)

**S3 Table. Associations of blood pressure categories with mortality from cardiovascular diseases and its major subtypes.** Values are hazard ratios (95% confidence interval).
(DOCX)

**S4 Table. Associations of prehypertension and hypertension subtypes with mortality from cardiovascular diseases and its major subtypes by sex.**
(DOCX)

**S5 Table. Associations of prehypertension and hypertension subtypes with mortality from cardiovascular diseases by survey sites.**
(DOCX)

**S6 Table. Associations of blood pressure categories with deaths due to cardiovascular diseases among participants excluding the first two years of follow-up.**
(DOCX)

**S7 Table. Associations of blood pressure categories with deaths of cardiovascular diseases among non-diabetes participants at baseline.**
(DOCX)

**S1 Text. Baseline questionnaire in the China Kadoorie Biobank study.**
(DOCX)

## Acknowledgments

The chief acknowledgment is to the participants, the project staff, and the China National Centre for Disease Control and Prevention (CDC) and its regional offices for assisting with the field-work. We thank Judith Mackay in Hong Kong; Yu Wang, Gonghuan Yang, Zhengfu Qiang, Lin Feng, Maigeng Zhou, Wenhua Zhao, and Yan Zhang in China CDC; Lingzhi Kong, Xiucheng Yu, and Kun Li in the Chinese Ministry of Health; and Sarah Clark, Martin Radley, Mike Hill, Hongchao Pan, and Jill Boreham in the CTSU, Oxford, for assisting with the design, planning, organization, and conduct of the study. Members of the China Kadoorie Biobank collaborative group as follows. International Steering Committee: Junshi Chen, Zhengming Chen (PI, E-mail: zhengming.chen@ctsu.ox.ac.uk), Robert Clarke, Rory Collins, Yu Guo, Liming Li (PI, E-mail: lmleeph@vip.163.com), Jun Lv, Richard Peto, Robin Walters. International Co-ordinating Centre, Oxford: Daniel Avery, Ruth Boxall, Derrick Bennett, Yumei Chang, Yiping Chen, Zhengming Chen, Robert Clarke, Huaidong Du, Simon Gilbert, Alex Hacker, Mike Hill, Michael Holmes, Andri Iona, Christiana Kartsonaki, Rene Kerosi, Ling Kong, Om Kurmi, Garry Lancaster, Sarah Lewington, Kuang Lin, John McDonnell, Iona Millwood, Qunhua Nie, Jayakrishnan Radhakrishnan, Paul Ryder, Sam Sansome, Dan Schmidt, Paul Sherliker, Rajani Sohoni, Becky Stevens, Iain Turnbull, Robin Walters, Jenny Wang, Lin Wang, Neil Wright, Ling Yang, Xiaoming Yang. National Co-ordinating Centre, Beijing: Zheng Bian, Yu Guo, Xiao Han, Can Hou, Jun Lv, Pei Pei, Chao Liu, Canqing Yu. 10 Regional Co-ordinating Centres: Qingdao CDC: Zengchang Pang, Ruqin Gao, Shanpeng Li, Shaojie Wang, Yongmei Liu, Ranran Du, Yajing Zang, Liang Cheng, Xiaocao Tian, Hua Zhang, Yaoming Zhai, Feng Ning, Xiaohui Sun, Feifei Li. Licang CDC: Silu Lv, Junzheng Wang, Wei Hou. Heilongjiang Provincial CDC: Mingyuan Zeng, Ge Jiang, Xue Zhou. Nangang CDC: Liqiu Yang, Hui He, Bo Yu, Yanjie Li, Qinai Xu, Quan Kang, Ziyan Guo. Hainan Provincial CDC: Dan Wang, Ximin Hu, Jinyan Chen, Yan Fu, Zhenwang Fu, Xiaohuan Wang. Meilan CDC: Min Weng, Zhendong Guo, Shukuan Wu, Yilei Li, Huimei Li, Zhifang Fu. Jiangsu Provincial CDC: Ming Wu, Yonglin Zhou, Jinyi Zhou, Ran Tao, Jie Yang, Jian Su. Suzhou CDC: Fang liu, Jun Zhang, Yihe Hu, Yan Lu, Liangcai Ma, Aiyu Tang, Shuo Zhang, Jianrong Jin, Jingchao Liu. Guangxi Provincial CDC: Zhenzhu Tang, Naying Chen, Ying Huang. Liuzhou CDC: Mingqiang Li, Jinhuai Meng, Rong Pan, Qilian Jiang, Jian Lan, Yun Liu, Liuping Wei, Liyuan Zhou, Ningyu Chen Ping Wang, Fanwen Meng, Yulu Qin, Sisi Wang. Sichuan Provincial CDC: Xianping Wu, Ningmei Zhang, Xiaofang Chen, Weiwei Zhou. Pengzhou CDC: Guojin Luo, Jianguo Li, Xiaofang Chen, Xunfu Zhong, Jiaqiu Liu, Qiang Sun. Gansu Provincial CDC: Pengfei Ge, Xiaolan Ren, Caixia Dong. Maiji CDC: Hui Zhang, Enke Mao, Xiaoping Wang, Tao Wang, Xi zhang. Henan Provincial CDC: Ding Zhang, Gang Zhou, Shixian Feng, Liang Chang, Lei Fan. Huixian CDC: Yulian Gao, Tianyou He, Huarong Sun, Pan He, Chen Hu, Xukui Zhang, Huifang Wu, Pan He. Zhejiang Provincial CDC: Min Yu, Ruying Hu, Hao Wang. Tongxiang CDC: Yijian Qian, Chunmei Wang, Kaixu Xie, Lingli Chen, Yidan Zhang, Dongxia Pan, Qijun Gu. Hunan Provincial CDC: Yuelong Huang, Biyun Chen, Li Yin, Huilin Liu, Zhongxi Fu, Qiaohua Xu. Liuyang CDC: Xin Xu, Hao Zhang, Huajun Long, Xianzhi Li, Libo Zhang, Zhe Qiu.

## Author Contributions

**Conceptualization:** Jie Guo, Jun Lv, Yiping Chen, Junshi Chen, Zhengming Chen, Canqing Yu, Liming Li.

**Data curation:** Yu Guo, Zheng Bian, Ling Yang, Yiping Chen, Jianqiang Zhang, Jvying Yao, Zhengming Chen, Liming Li.

**Formal analysis:** Jie Guo, Bang Zheng, Man Wu, Ling Yang, Yiping Chen, Jian Su, Jianqiang Zhang, Jvying Yao, Junshi Chen, Zhengming Chen.

**Funding acquisition:** Jun Lv, Liming Li.

**Investigation:** Jie Guo, Jun Lv, Yu Guo, Zheng Bian, Jian Su, Jianqiang Zhang, Jvying Yao, Junshi Chen, Zhengming Chen, Canqing Yu.

**Methodology:** Jie Guo, Bang Zheng, Man Wu, Jian Su, Canqing Yu.

**Project administration:** Yu Guo, Zheng Bian, Canqing Yu, Liming Li.

**Resources:** Canqing Yu.

**Software:** Jie Guo, Bang Zheng.

**Supervision:** Canqing Yu, Liming Li.

**Validation:** Jun Lv, Bang Zheng, Man Wu, Ling Yang, Junshi Chen, Zhengming Chen, Canqing Yu, Liming Li.

**Writing – original draft:** Jie Guo, Jun Lv.

**Writing – review & editing:** Jie Guo, Jun Lv, Yu Guo, Zheng Bian, Bang Zheng, Man Wu, Ling Yang, Yiping Chen, Jian Su, Jianqiang Zhang, Jvying Yao, Junshi Chen, Zhengming Chen, Canqing Yu, Liming Li.

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
