## [Decision Letter · Decision Letter 0]

12 Apr 2021

PONE-D-21-07340

Association between Blood Pressure Categories and Cardiovascular Disease Mortality in China

PLOS ONE

Dear Dr. Yu,

Thank you for submitting your manuscript to PLOS ONE. After careful consideration, we feel that it has merit but does not fully meet PLOS ONE’s publication criteria as it currently stands. Therefore, we invite you to submit a revised version of the manuscript that addresses the points raised during the review process.

Please revise the manuscript according to the comments of the three Reviewers. In addition, please consider analyses using blood pressure as a continuous variable.

We look forward to receiving your revised manuscript.

Kind regards,

Yan Li, MD, PhD

Academic Editor

PLOS ONE

Journal Requirements:

3. One of the noted authors is a group or consortium; China Kadoorie Biobank Collaborative Group.

In addition to naming the author group, please list the individual authors and affiliations within this group in the acknowledgments section of your manuscript.

Please also indicate clearly a lead author for this group along with a contact email address.

Reviewers' comments:

Reviewer's Responses to Questions

**Comments to the Author**

1. Is the manuscript technically sound, and do the data support the conclusions?

Reviewer #1: Yes

Reviewer #2: Partly

Reviewer #3: Yes

2. Has the statistical analysis been performed appropriately and rigorously? 

Reviewer #1: Yes

Reviewer #2: Yes

Reviewer #3: Yes

3. Have the authors made all data underlying the findings in their manuscript fully available?

Reviewer #1: No

Reviewer #2: Yes

Reviewer #3: Yes

4. Is the manuscript presented in an intelligible fashion and written in standard English?

Reviewer #1: Yes

Reviewer #2: Yes

Reviewer #3: Yes

5. Review Comments to the Author

Reviewer #1: The authors' work with large prospective cohort (n=431,000, 10-year follow-up) provided information on blood pressure (BP) category and prospective mortality risk from cardiovascular (CV) origin. Importantly, prehypertension, along with all subcategories of hypertension, was associated with increased mortality compared to normotensive participants in adult Chinese free of baseline antihypertensive drug, CV disease or cancer.

Major comments

1. High BP could aggravate other non-CV diseases. The hazard ratio (HR) of BP categories to all-cause and non-CV mortality should therefore be reported. CV death, as a competing event to death from other causes, should be evaluated by competing risk regression.

2. There could be discrepancies on the mortality rate and HR of BP categories with mortality among the included centers. Center-specific statistics should as well be provided.

Minor comments

1. The limitation part in the abstract (line 56-58), albeit modestly written, could lead to a suspicion in the first impression on the credibility of results and conclusion. In the abstract, it's better to put "BP was measured twice in a single visit" into the methods section and the limitation should be omitted.

2. Following the line, in methods part, line 119-120, the authors should rephrase the measurement of blood pressure as "occasion" and "reading" are two distinct entities. The authors may refer to their Lacey 2018 Lancet Global Health paper (PMID 29773120) for more precise description on the method of BP measurement.

3. Standard Error SE used in Table 1 is inversely correlated to sample size, and therefore definitely small when the sample size is huge. Change it to SD to show the distribution of continuous variables.

4. line 88: "the number of hypertension" should be the prevalence of hypertension or the number of participants with hypertension.

5. line 96: there is limited evidence for the associations of 'BP categories' with the....

6. line 113: Check if the study was performed in accordance with the Declaration of Helsinki, and add to the end of this paragraph.

Reviewer #2: “Association between Blood Pressure Categories and Cardiovascular Disease Mortality in China” divided the population into 6 categories based on the systolic and diastolic blood pressure levels among approximately 430,000 general people without antihypertensive drugs or medical history of cardiovascular disease. The study found that compared with people with normal blood pressure, the other 5 categories were all related to compound cardiovascular death after adjustment. At the same time, the study found that cerebrovascular mortality was most closely related to blood pressure categories in the Chinese population. Besides, age was a confounding factor that affected the strength of the association between blood pressure categories and cardiovascular death.

The highlight of this study, as the author mentioned, the sample size was very large, and the study covered 10 regions in China.

There however remain some points that might be clarified:

1) In the methodology (lines 108-109), please add that this study was a general population study when you briefly described the CKB study. Otherwise, it will be liable to cause misunderstandings.

2) The average blood pressure used in this study is the average of the last two blood pressure measurements. Will there be different results and conclusions if the average of three consecutive blood pressure measurements is taken?

3) This study only analyzed the relationship between cardiovascular death and blood pressure categories. Why is there no further analysis of all-cause death? If feasible, please add results and discussion about the relationship between all-cause death and blood pressure categories.

4) The statistical description of the constituent ratio in Figure 1 is not visualized enough using histograms, and it is recommended to modify it to a pie chart.

5) When the continuous variables in Table 1 were used for statistical description, should the standard deviation rather than the standard error be considered for the degree of dispersion?

6) The incidence rate of hemorrhagic stroke in Table 2 is higher than that of ischemic stroke. Is it different from the constituent ratio of stroke types in the Chinese population? Please explain further.

7) The study concluded that prehypertension-low was associated with higher cardiovascular deaths. However, in Table S2, prehypertension-low is actually only associated with hemorrhagic stroke, HR1.26 (1.07-1.48), and other single-factor deaths were not related to prehypertension-low. Is there more sufficient evidence to make such an extrapolated conclusion cautiously?

8) In Table S2, the HR of most blood pressure types in Model 1, Model 2 and Model 3 were gradually increasing. In the Cox risk regression model, as the number of independent variables added, the contribution of the research factors of interest to the outcome generally decreases. This phenomenon was confusing, please explain further.

Reviewer #3: This study, involving 0.4 million people, investigated the association between hypertension subtypes and CVD outcomes. It was well written.

There are several comments that might be taken into consideration.

1. It would be better to demonstrated the “inclusion criteria” and “exclusion criteria” of the CKB study.

2. For better understanding, it is recommended to use the definition of hypertension grades according to the 2018 ESC guidelines of hypertension as “optimal, normal, high normal”, rather than using the terms as “normal, prehypertension-low, prehypertension-high”.

3. Please state why patients with antihypertension treatment at baseline were excluded.

4. Please state the proportion of “loss to follow-up” and how it was addressed.

6. PLOS authors have the option to publish the peer review history of their article (what does this mean?). If published, this will include your full peer review and any attached files.

Reviewer #1: No

Reviewer #2: No

Reviewer #3: **Yes: **Kun Xie

---

## [Author Response · Author response to Decision Letter 0]

10 May 2021

Response to Reviewers’ comments 

Ms. Ref. No.: PONE-D-21-07340

We thank the editor’s help and we have revised the manuscript based on the PLOS ONE’S requirements carefully.

We thank the editor’s help, and we would like to revise the Data Availability statement as follows.

“The authors do not own the data for this study. Requests for the data should be submitted to the China Kadoorie Biobank (CKB) Data Access Committee (http://www.ckbiobank.org/site/Research/Data+Access+Policy). As stated in the policy, as data custodian, the CKB study group must maintain the integrity of the database for future use and regulate data access to comply with prior conditions agreed with the Chinese government. Data security is an integral part of CKB study protocols. Data can be released outside the CKB research group only with appropriate security safeguards.”

3. One of the noted authors is a group or consortium; China Kadoorie Biobank Collaborative Group. In addition to naming the author group, please list the individual authors and affiliations within this group in the acknowledgments section of your manuscript. Please also indicate clearly a lead author for this group along with a contact email address.

Following the editor’s suggestion, we added the members of the China Kadoorie Biobank collaborative group in the acknowledgements (pages 24-25, lines 392-429).

 

Please revise the manuscript according to the comments of the three Reviewers. In addition, please consider analyses using blood pressure as a continuous variable.

We thank the reviewers for their valuable comments and thus for the improvement of our paper. All comments and suggestions have been carefully addressed in the revised manuscript. Details of the comments are responded to point-by-point as follows. One previous paper based on CKB study had explored the association between blood pressure as a continuous variable and CVD (Lacey Ben, et al, The Lancet Global Health, 2018), so in this manuscript, we mainly focused on the risk effect of blood pressure categories and aimed to provide more evidence about the newly defined hypertension in 2017 ACC/AHA and hypertension subtypes.

Reviewer #1: The authors' work with large prospective cohort (n=431,000, 10-year follow-up) provided information on blood pressure (BP) category and prospective mortality risk from cardiovascular (CV) origin. Importantly, prehypertension, along with all subcategories of hypertension, was associated with increased mortality compared to normotensive participants in adult Chinese free of baseline antihypertensive drug, CV disease or cancer.

Major comments

1. High BP could aggravate other non-CV diseases. The hazard ratio (HR) of BP categories to all-cause and non-CV mortality should therefore be reported. CV death, as a competing event to death from other causes, should be evaluated by competing risk regression.

Thank the review’s comments. Considering that the burden of high BP on the health system is mainly through CVDs and the heavy burden of CVD, especially cerebrovascular diseases, in the Chinese population, we mainly focused on exploring the effect of BP categories on CVD and CVD subtypes in the current study. We agree that the associations of BP categories with all-cause and non-CVD mortality are important considerations but not within the scope of this study. In the future study, we would explore the role of BP in the development of all-cause and non-CVD mortality.

2. There could be discrepancies on the mortality rate and HR of BP categories with mortality among the included centers. Center-specific statistics should as well be provided.

We thank the reviewer’s comments. We agreed with the reviewer that there might be discrepancies in the associations between BP categories and CVD mortality by survey sites. To deal with this issue, we conducted the Cox regression models with stratification according to survey sites (10 regions), as well as age at risk (in 5-year intervals) and sex.

Moreover, we conducted subgroup analyses by 10 regions and added the results in the supplementary file (S5 Table) and in the manuscript (page 18, lines 277-279) as follows.

“S5 Table shows the associations between BP categories and CVD mortality across 10 survey sites. There was a statistically significant interaction between BP categories and survey sites (P for interaction = 0.016).”

Minor comments

1. The limitation part in the abstract (line 56-58), albeit modestly written, could lead to a suspicion in the first impression on the credibility of results and conclusion. In the abstract, it's better to put "BP was measured twice in a single visit" into the methods section and the limitation should be omitted.

We thank the reviewer for pointing this out. Following the reviewer’s suggestion, we revised the methods (page 3, lines 46-47) as follows “BP was measured twice in a single visit at baseline”. As the reviewer commented, to avoid confusion, we delete the limitation in the abstract.

2. Following the line, in methods part, line 119-120, the authors should rephrase the measurement of blood pressure as "occasion" and "reading" are two distinct entities. The authors may refer to their Lacey 2018 Lancet Global Health paper (PMID 29773120) for more precise description on the method of BP measurement.

We appreciate the reviewer’s careful reading, and we have corrected it (page 7, lines 122-127) as follows. “BP was measured twice by trained staff using a UA-779 digital monitor after they remained at rest in the seated position for at least 5 minutes (Longo D et al, Blood Press Monit 2002). If the difference between the two measurements was >10 mm Hg for SBP, a third measurement was required. Only the last two readings were recorded and used to calculate the average of SBP and DBP (Lacey B et al, Lancet Global Health 2018).”

3. Standard Error SE used in Table 1 is inversely correlated to sample size, and therefore definitely small when the sample size is huge. Change it to SD to show the distribution of continuous variables.

We thank the reviewer for the comments. Considering the potential effect of age, sex, and survey sites for the distribution of baseline characteristics across BP categories, we conducted linear regression with adjustment for age, sex, and survey sites to analyse the distribution of continuous variables. Thus, in Table 1, we showed the mean and standard error derived from the adjusted linear model. 

Following the reviewer’s suggestion, we added one table (S1 Table) in the supplementary file to describe the crude means (standard deviation) and percentages and revised the corresponding content in the manuscript (page 12, lines 229-230).

“The distribution of baseline characteristics of the study population without adjustment for age, sex, and survey sites was presented in S1 Table.”

4. line 88: "the number of hypertension" should be the prevalence of hypertension or the number of participants with hypertension.

Thank the reviewer for the correction. We have corrected the term (page 5, line 91) as follows. “…the prevalence of hypertension…”.

5. line 96: there is limited evidence for the associations of 'BP categories' with the....

Following the reviewer’s suggestion, we revised the sentence (page 5, lines 99-100) as follows. “there is limited evidence for the associations of BP categories with the major subtypes of cardiovascular disease…”

6. line 113: Check if the study was performed in accordance with the Declaration of Helsinki, and add to the end of this paragraph.

As the reviewer commented, we added one sentence in the Study population (page 6, line 116). “The study was in accordance with Declaration of Helsinki.” 

Reviewer #2: “Association between Blood Pressure Categories and Cardiovascular Disease Mortality in China” divided the population into 6 categories based on the systolic and diastolic blood pressure levels among approximately 430,000 general people without antihypertensive drugs or medical history of cardiovascular disease. The study found that compared with people with normal blood pressure, the other 5 categories were all related to compound cardiovascular death after adjustment. At the same time, the study found that cerebrovascular mortality was most closely related to blood pressure categories in the Chinese population. Besides, age was a confounding factor that affected the strength of the association between blood pressure categories and cardiovascular death.

The highlight of this study, as the author mentioned, the sample size was very large, and the study covered 10 regions in China.

There however remain some points that might be clarified:

1) In the methodology (lines 108-109), please add that this study was a general population study when you briefly described the CKB study. Otherwise, it will be liable to cause misunderstandings.

Thank the reviewer for pointing this out. We have revised this part (page 6, line 111) to avoid misunderstanding.

“The CKB is a community-based prospective cohort study, involving over 0.5 million adults aged 30-79 years in 10 areas of China between 2004 and 2008.”

2) The average blood pressure used in this study is the average of the last two blood pressure measurements. Will there be different results and conclusions if the average of three consecutive blood pressure measurements is taken?

If the difference between the two measurements was >10 mm Hg for SBP, a third measurement was required. Only the last two readings were recorded so it was not possible to calculate the average BP using three consecutive measurements. To avoid confusion, we have rephrased the assessment of BP as follows (page 7, lines 132-133).

“Only the last two readings were recorded and used to calculate the average of SBP and DBP (Lacey B et al, Lancet Global Health 2018).”

3) This study only analyzed the relationship between cardiovascular death and blood pressure categories. Why is there no further analysis of all-cause death? If feasible, please add results and discussion about the relationship between all-cause death and blood pressure categories.

Thank the review’s comments. Considering that the burden of high BP on the health system is mainly through CVDs and the heavy burden of CVD, especially cerebrovascular diseases, in the Chinese population, we mainly focused on exploring the effect of BP categories on CVD and CVD subtypes in the current study. Moreover, we want to provide more evidence about the effect of newly defined hypertension in the 2017 ACC/AHA guideline given that previous studies showed mixed results about its effect on CVD mortality and little evidence is available for its effect on CVD subtypes. Thus, in the current study, we focused on CVD but not all-cause death. Future studies would be conducted to explore the associations of BP with all-cause and non-CVD mortality.

4) The statistical description of the constituent ratio in Figure 1 is not visualized enough using histograms, and it is recommended to modify it to a pie chart.

Thank the reviewer’s suggestion. Besides showing the proportions of different BP categories, we also want to present the trend of proportions of different BP categories with aging. Therefore, we decided to use the histogram graph to visualise and compare the height of the corresponding bar across age groups.

5) When the continuous variables in Table 1 were used for statistical description, should the standard deviation rather than the standard error be considered for the degree of dispersion?

Thank the review for pointing this out. Considering the potential effect of age, sex, and survey sites for the distribution of baseline characteristics across BP categories, we conducted linear regression with adjustment for age, sex, and survey sites to analyse the distribution of continuous variables. Thus, in Table 1, we showed the mean and standard error derived from the adjusted linear model. We added one table (S1 Table) in supplementary file to describe the crude means and standard deviations (page 12, lines 229-230).

“The distribution of baseline characteristics of the study population without adjustment for age, sex, and survey sites was presented in S1 Table.”

6) The incidence rate of hemorrhagic stroke in Table 2 is higher than that of ischemic stroke. Is it different from the constituent ratio of stroke types in the Chinese population? Please explain further.

Thank the reviewer’s comment. In the current study, the mortality of hemorrhagic stroke is higher than that of ischemic stroke, which is consistent with previous studies conducted among a Chinese population (Yang G, et al, Lancet 2013; Zhou M, et al, Lancet 2016). Zhou M et al reported that the age-standardised death rate per 100 000 people was 62.9 for ischemic stroke vs. 94.4 for hemorrhagic stroke in 2013 among a Chinese population. We added the previous evidence in the manuscript (page 6, lines 99-100).

“…and the mortality from hemorrhagic stroke was higher than that from ischemic stroke (Zhou M, et al, Lancet 2016)”

7) The study concluded that prehypertension-low was associated with higher cardiovascular deaths. However, in Table S2, prehypertension-low is actually only associated with hemorrhagic stroke, HR1.26 (1.07-1.48), and other single-factor deaths were not related to prehypertension-low. Is there more sufficient evidence to make such an extrapolated conclusion cautiously?

We understand the reviewer’s concern. Our results showed that prehypertension-low was associated with the increased risk of death from hemorrhagic stroke. There were no statistically significant associations between prehypertension-low and mortality from other CVD subtypes. Given that the analyses were conducted with a sufficient sample size and the point estimates of hazard risk for other CVD subtypes were around 1, we are confident about the results that prehypertension-low was not associated with increased CVD risk except for hemorrhagic stroke. However, for prehypertension-high (newly defined as hypertension in 2017 ACC/AHA guideline), it related to all subtypes of CVDs, so we concluded that it should be paid attention to in the primary prevention of CVD.

8) In Table S2, the HR of most blood pressure types in Model 1, Model 2 and Model 3 were gradually increasing. In the Cox risk regression model, as the number of independent variables added, the contribution of the research factors of interest to the outcome generally decreases. This phenomenon was confusing, please explain further.

In the Table S2, we presented the associations using the Cox regression model with adjustment for covariates or step by step to explore whether these potential confounders could change the associations. A confounder could be a risk factor (e.g., age), a preventive factor (e.g., being physically active), or a surrogate or a marker of other causes (e.g., marital status and education level) for the CVD outcomes. Thus, confounders might distort the estimate of association in one direction or another. For example, a family history of corresponding diseases might increase the risk of disease, but participants with a family history might be more likely to improve their lifestyles which could further decrease the risk. Moreover, after adjustment, both the point estimates and the confidence intervals did not change substantially. 

 

Reviewer #3: This study, involving 0.4 million people, investigated the association between hypertension subtypes and CVD outcomes. It was well written.

There are several comments that might be taken into consideration.

1. It would be better to demonstrated the “inclusion criteria” and “exclusion criteria” of the CKB study.

We thank the reviewer’s suggestions. We added the inclusion criteria and exclusion criteria in the manuscript as follows (page 6, lines 204-206).

“All men and women aged 30-79 years who were permanently resident and without major disability in each administrative unit were identified and invited to participate (Chen Z, et al. Int J Epidemiol 2005).”

2. For better understanding, it is recommended to use the definition of hypertension grades according to the 2018 ESC guidelines of hypertension as “optimal, normal, high normal”, rather than using the terms as “normal, prehypertension-low, prehypertension-high”.

We thank the reviewer’s suggestion. In the current study, we further stratified the “prehypertension” into two groups (SBP 120−129 and DBP <80 mm Hg; SBP 130−139 and/or DBP 80−89 mm Hg) using the criteria from the 2017 ACC/AHA hypertension guideline to estimate the risk of the newly defined “Hypertension stage 1”. The cut-off for the “Hypertension stage 1” (SBP 130−139 and/or DBP 80−89 mm Hg) is different from the 2018 ESC guidelines (High-normal BP: SBP 130−139 and/or DBP 85−89 mm Hg). Thus, in the manuscript, we used the terms “prehypertension-low and prehypertension-high” to name those categories instead of using the terms from 2018 ESC guidelines.

3. Please state why patients with antihypertension treatment at baseline were excluded.

Thank the reviewer’s comment. We have stated the reason in the manuscript (page 7, lines 123-125) as follows.

“Moreover, we further excluded participants with the antihypertension treatment at baseline (n=65 168), because both dose and types of antihypertensive medications may influence BP levels and lead to misclassification of BP categories.”

4. Please state the proportion of “loss to follow-up” and how it was addressed.

By December 31, 2016, 4434 (1.03%) participants were lost to follow-up. In the analyses, if participants were lost to follow-up without information about death, the survival time was calculated from baseline to the date of the latest follow-up date. We stated the rate of loss to follow-up in the manuscript as follows (page 7, line 126; page 9, lines 176-177).

“By December 31, 2016, 4434 (1.03%) participants were lost to follow-up.”

“For the current study, survival time was calculated from baseline to the date of death, loss to follow-up, or 31 December 2016, whichever occurred first.”

---

## [Decision Letter · Decision Letter 1]

3 Jun 2021

PONE-D-21-07340R1

Association between blood pressure categories and cardiovascular disease mortality in China

PLOS ONE

Dear Dr. Yu,

Thank you for submitting your manuscript to PLOS ONE. After careful consideration, we feel that it has merit but does not fully meet PLOS ONE’s publication criteria as it currently stands. Therefore, we invite you to submit a revised version of the manuscript that addresses the points raised during the review process. 

Please revise the manuscript according to Reviewer 3's minor comments. 

We look forward to receiving your revised manuscript.

Kind regards,

Yan Li, MD, PhD

Academic Editor

PLOS ONE

Journal Requirements:

Reviewers' comments:

Reviewer's Responses to Questions

**Comments to the Author**

1. If the authors have adequately addressed your comments raised in a previous round of review and you feel that this manuscript is now acceptable for publication, you may indicate that here to bypass the “Comments to the Author” section, enter your conflict of interest statement in the “Confidential to Editor” section, and submit your "Accept" recommendation.

Reviewer #1: All comments have been addressed

Reviewer #2: All comments have been addressed

Reviewer #3: All comments have been addressed

2. Is the manuscript technically sound, and do the data support the conclusions?

Reviewer #1: (No Response)

Reviewer #2: Yes

Reviewer #3: Yes

3. Has the statistical analysis been performed appropriately and rigorously? 

Reviewer #1: (No Response)

Reviewer #2: Yes

Reviewer #3: Yes

4. Have the authors made all data underlying the findings in their manuscript fully available?

Reviewer #1: (No Response)

Reviewer #2: Yes

Reviewer #3: No

5. Is the manuscript presented in an intelligible fashion and written in standard English?

Reviewer #1: (No Response)

Reviewer #2: Yes

Reviewer #3: Yes

6. Review Comments to the Author

Reviewer #1: (No Response)

Reviewer #2: (No Response)

Reviewer #3: There are two minor comments which need to be addressed.

1. It would cause selection bias if those patients with anti-hypertensive agents were excluded. In the current study, anti-hypertensive treatment could be an unmeasured covariate which would affect the results. It should be further discussed in the part of Discussion.

2. It’s a little bit confusing that the author defined “prehypertension-high” as “hypertension stage 1” according to ACC/AHA guideline, while the definitions of IDH and ISH were not consist with ACC/AHA guideline. It should have a much clearer expression.

7. PLOS authors have the option to publish the peer review history of their article (what does this mean?). If published, this will include your full peer review and any attached files.

Reviewer #1: No

Reviewer #2: No

Reviewer #3: No

---

## [Author Response · Author response to Decision Letter 1]

14 Jun 2021

There are two minor comments which need to be addressed.

We thank the reviewer for the valuable comments. All comments and suggestions have been carefully addressed in the revised manuscript. Details of the comments are responded to point-by-point as follows.

1. It would cause selection bias if those patients with antihypertensive agents were excluded. In the current study, antihypertensive treatment could be an unmeasured covariate which would affect the results. It should be further discussed in the part of Discussion.

We thank the reviewer's comments. We agree that excluding patients with antihypertension agents might cause selection bias given that hypertension with treatment might be more aware of their health status and more accessible to health care or might have severe symptom due to high blood pressure compared to those without treatment, which could affect future CVD risk. However, taking antihypertensive medicines would change the level of blood pressure and lead to a misclassification of hypertension subtypes. Given that it is not possible to know their blood pressure level before taking antihypertensive medicines, to exclude participants with antihypertensive medicines is more reasonable for the current study. Thus, our findings could be generalized to the population without taking antihypertensive medicines.

We added the corresponding issue in the limitations section of discussion (page 21, lines 359-364).

"Second, we excluded participants who taking antihypertensive medicines, which might cause selection bias and limit the extrapolation of our findings. However, antihypertensive medication would affect the patients' blood pressure level, leading to misclassification of BP categories. Hence, it is more reasonable to restrict study participants without antihypertensive treatment."

2. It's a little bit confusing that the author defined "prehypertension-high" as "hypertension stage 1" according to ACC/AHA guideline, while the definitions of IDH and ISH were not consist with ACC/AHA guideline. It should have a much clearer expression.

We thank the reviewer's comments. In the current study, we defined the blood pressure categories mainly referred to the JNC-7 guideline, which defined prehypertension and hypertension subtypes (i.e. ISH, IDH, and SDH). The definition was adopted in previous studies (Arima H et al, Hypertension 2012; Kelly TN et al, Circulation 2008). However, the prehypertension (i.e., SBP/DBP 120 to 139/80 to 89 mm Hg) had a broad range of blood pressure, and its effects on CVD risk were heterogeneous within prehypertension (Y Huang et al, American Heart Journal 2014). In addition, the 2017 ACC/AHA guideline newly defined the high-range blood pressure within prehypertension as hypertension. Thus, we combined the prehypertension definition in JNC-7 with the new cut-off in 2017 ACC/AHA guideline, i.e., SBP 120−129 and DBP <80 mm Hg; SBP 130−139 and/or DBP 80−89 mm Hg. We revised the manuscript (page 7, lines 127-132) to clarify our categorisation.

"According to the JNC-7, BP categories were defined into five groups 1) normal (SBP <120 and DBP <80 mm Hg); 2) prehypertension (SBP 120−139 and/or DBP 80−89 mm Hg); 3) ISH (SBP ≥140 and DBP <90 mm Hg); 4) IDH (SBP <140 and DBP ≥90 mm Hg); 5) SDH (SBP ≥140 and DBP ≥90 mm Hg) (Chobanian A V., Hypertension 2003). In the 2017 ACC/AHA hypertension guideline, hypertension was defined as SBP ≥130 mmHg and/or DPB ≥90 mmHg (Whelton PK, Hypertension 2018). To estimate the effect of "Elevated" and "Hypertension stage 1", we further divided prehypertension into prehypertension-low (equal to "Elevated", SBP 120−129 and DBP <80 mm Hg) and prehypertension-high (equal to "Hypertension stage 1", SBP 130−139 and/or DBP 80−89 mm Hg) (Whelton PK, Hypertension 2018)."

Reviewer's Responses to Questions

Have the authors made all data underlying the findings in their manuscript fully available?

Reviewer #1: (No Response)

Reviewer #2: Yes

Reviewer #3: No

For the data availability, we want to state as follows.

"According to the Regulation of the People's Republic of China on the Administration of Human Genetic Resources, we are not allowed to provide Chinese human clinical and genetic data abroad without official approval so the data in the present study cannot be shared without restrictions. However, researchers who are interested in accessing and analyzing data collected in the China Kadoorie Biobank (CKB) study may contact the data use and access committee (http://www.ckbiobank.org/site/Research/Data+Access+Policy). As stated in the policy, as data custodian, the CKB study group must maintain the integrity of the database for future use and regulate data access to comply with prior conditions agreed with the Chinese government."

---

## [Decision Letter · Decision Letter 2]

15 Jul 2021

Association between blood pressure categories and cardiovascular disease mortality in China

PONE-D-21-07340R2

Dear Dr. Yu,

We’re pleased to inform you that your manuscript has been judged scientifically suitable for publication and will be formally accepted for publication once it meets all outstanding technical requirements.

Kind regards,

Yan Li, MD, PhD

Academic Editor

PLOS ONE

Additional Editor Comments (optional):

Reviewers' comments:

Reviewer's Responses to Questions

**Comments to the Author**

1. If the authors have adequately addressed your comments raised in a previous round of review and you feel that this manuscript is now acceptable for publication, you may indicate that here to bypass the “Comments to the Author” section, enter your conflict of interest statement in the “Confidential to Editor” section, and submit your "Accept" recommendation.

Reviewer #1: All comments have been addressed

Reviewer #2: All comments have been addressed

Reviewer #3: All comments have been addressed

2. Is the manuscript technically sound, and do the data support the conclusions?

Reviewer #1: (No Response)

Reviewer #2: Yes

Reviewer #3: Yes

3. Has the statistical analysis been performed appropriately and rigorously? 

Reviewer #1: (No Response)

Reviewer #2: Yes

Reviewer #3: Yes

4. Have the authors made all data underlying the findings in their manuscript fully available?

Reviewer #1: (No Response)

Reviewer #2: Yes

Reviewer #3: Yes

5. Is the manuscript presented in an intelligible fashion and written in standard English?

Reviewer #1: (No Response)

Reviewer #2: Yes

Reviewer #3: Yes

6. Review Comments to the Author

Reviewer #1: (No Response)

Reviewer #2: (No Response)

Reviewer #3: This study reported CVD endpoints among 0.5 million adults in primary prevention in China. It was well written and should be accepted.

7. PLOS authors have the option to publish the peer review history of their article (what does this mean?). If published, this will include your full peer review and any attached files.

Reviewer #1: No

Reviewer #2: No

Reviewer #3: No

---

## [Editor Report · Acceptance letter]

19 Jul 2021

PONE-D-21-07340R2 

Association between blood pressure categories and cardiovascular disease mortality in China 

Dear Dr. Yu:

I'm pleased to inform you that your manuscript has been deemed suitable for publication in PLOS ONE. Congratulations! Your manuscript is now with our production department. 

Kind regards, 

on behalf of

Professor Yan Li 

Academic Editor

PLOS ONE